# The effect of a family-centered advance care planning intervention for persons with cognitive impairment and their family caregivers on end-of-life care discussions and decisions

**Hsiu-Li Huang** [1]*, **Wei-Ru Lu**[2], **Huei-Ling Huang**[3], **Chien-Liang Liu**[4]

1 Department of Long-Term Care, College of Health Technology, National Taipei University of Nursing and Health Sciences, Taipei, Taiwan, 2 Department of Nursing, Sijhih Cathay General Hospital, New Taipei City, Taiwan, 3 Department of Gerontology and Health Care Management, Chang Gung University of Science and Technology, Taoyuan, Taiwan, 4 Dementia Center, Department of Internal Medicine, Heping Fuyou Branch, Taipei City Hospital, Taipei, Taiwan

* hsiuli@ntunhs.edu.tw

**Data Availability Statement:** All S1 files are available from the S1 Dataset.

## Abstract

Advanced care planning (ACP) includes advance directives (AD), which can specify provisions for palliative care and types of life-sustaining treatments for an individual requiring end-of-life (EoL) care. ACP for persons in the early stages of cognitive decline can decrease anxiety and conflict for family members needing to make decisions about EoL-care, which is especially critical for family caregivers (FCGs) if they play a role as a surrogate regarding healthcare decisions. However, ACP for persons with cognitive impairment (PWCIs) is often overlooked. This study explored the effects of a family-centered ACP intervention on decisions about EoL-care, life-sustaining treatment decisions, and discussions of related topics among PWCIs and FCGs. The study was conducted in outpatient clinics of regional teaching hospitals in northern Taiwan. Participants were dyads consisting of persons diagnosed with mild cognitive impairment or mild dementia and their FCGs. The family-centered ACP intervention was provided by an ACP-trained senior registered nurse. A one-group, pretest–posttest design was used to evaluate the effect of the intervention on 44 dyads. Four structured questionnaires collected data regarding familiarity with ACP, intention to engage in ACP, participation in personal discussions between the dyads about ACP, and consistency between PWCIs and FCGs for decisions about life-sustaining treatments at EoL. Paired t, Kappa, and McNemar tests were used to compare differences between pre-intervention data (pretest) and post-intervention data (posttest). There were significant increases in familiarity with ACP, components of ACP, and the number of topics PWCIs and FCGs personally discussed surrounding EoL-care decisions. There was no change for either group in wanting to have a formal ACP consultation and only modest increases in consistency between PWCIs and FCGs for life-sustaining treatment decisions after completion of the family-centered ACP intervention. Clinicians caring for PWCIs should incorporate family-centered ACP interventions and support ongoing discussions about life-sustaining medical

**Funding:** This study received a fund from the Ministry of Science and Technology of Taiwan (MOST 106-2314-B-182 -006 -MY3) and (MOST 109-2314-B-227 -006 -MY3). The funder had no role in study design, data collection, and analysis, decision to publish, or preparation of the manuscript.

**Competing interests:** The authors have declared that no competing interests exist.

treatments to ensure their preferences regarding EoL-care are respected. The accessibility and availability of consultations about ACP should also be provided.

## 1. Introduction

Dementia is a progressive neurodegenerative disease that is life-limiting. An inevitable consequence of dementia is that the mental status of persons with cognitive impairment (PWCIs) will gradually deteriorate to the end-of-life (EoL) stage [1, 2]. Advanced dementia puts individuals at risk of inadequate EoL-care due to decreases in cognitive and decision-making abilities. Advance care planning (ACP) is one means of improving quality of EoL-care for PWCIs and their family caregivers (FCGs) [1, 3]. ACP is an umbrella term that includes personal, clinical, legal, and financial planning in preparation for the time an individual no longer has the capacity to make their own decisions about healthcare or legal proceedings. The process of ACP often includes advance directives (ADs), which are formal written documents describing preferences for future healthcare, including directives about do-not-resuscitate (DNR), intubation and artificial respiration, and tube feeding, as well as appointing a surrogate to make healthcare decisions in the event of incapacity [1, 2]. However, an individual can provide ADs without ACP, which is more extensive in terms of the number of personal preferences included.

ACP can be conducted formally through consultations with healthcare or legal professionals, or through informal discussions with family members [4]. A recent consensus paper defined ACP for health care as, '*the ability to enable individuals to define goals and preferences for future medical treatment and care, to discuss these goals and preferences with family and healthcare providers, and to record and review these preferences if appropriate*' [5]. National experts recommend that formal ACP should address the individual's concerns and values across the physical, psychological, social, and spiritual domains and to tailor holistic care that responds to the wishes of the individual [5].

The benefits of ACP include decreased hospitalizations, increased conformity between care received and a patient's wishes and improved outcomes for care at end-of-life (EoL) [3, 6, 7]. An ACP has also been shown to increase FCGs' satisfaction with a patient's EoL-care in addition to reducing stress, anxiety, depression, and family conflicts [3, 8–10]. However, internationally, ACP for PWCIs is not popular and less than 40% of PWCIs have are given the opportunity to participate in formal ACP [7, 11]. As the worldwide population of older adults continues to increase, the incidence of PWCIs is also increasing. Therefore, strategies to improve this population's access to information about ACP options should be a concern of healthcare providers, which could also improve EoL care for PWCIs.

There is a consensus among experts that ACP should be completed during the early stages of dementia, before an individual's ability to consider the future is compromised [1, 8, 12]. Factors affecting the completion of ACP include personal characteristics such as education and age, and cultural factors such as ethnicity, social expectations, and EoL-care policies [13]. Not only is ACP low for PWCIs, but the percent of the population with mild to moderate cognitive impairment that has completed an AD in Western countries ranges from 9% [14] to 87% [4]. Increasing participation in ACP for PWCIs requires different strategies, particularly for populations in Eastern countries where ACP is not a common practice.

In 2000, Taiwan enacted the first Hospice Palliative Care Act (HPCA) on self-determination of the terminally ill, which allowed palliative care to replace cardiopulmonary resuscitation (CPR) and other life-sustaining medical treatments during EoL. The individual is officially allowed to make an AD that is a "living will for hospice palliative care or do-not-

resuscitate (DNR)" or consent for DNR signed by a surrogate, but this does not require participation in an ACP consultation [15]. A person who demonstrates legal capacity may prewrite the aforementioned documents of AD under witness of two adults with full disposing capacity. The AD can be uploaded to the health insurance card system by a designated hospital or institution, without going through an ACP consultation. However, the HPCA only guarantees the right of medical decisions for terminally ill patients, which excludes patients with diseases that are incurable, painful, and difficult to judge as terminal.

A patient's dignity in life should also be protected, which includes providing an opportunity to decide on treatment options. Therefore, the Legislative Yuan passed the Patient Right to Autonomy Act (PRAA) in December 2015, which was officially implemented in January 2019. This Act once again declares that autonomous medical decisions are a universal human right. The PRAA expanded the target scope to include not only terminally ill patients, but also patients in an irreversible coma, a permanent vegetative state, suffering from severe dementia, and other disease conditions or states of suffering considered unbearable or incurable and for which no other appropriate treatment options are available. PRAA Article 9 reads, "To establish an advance directive, the declarant must fulfill the following requirements":

1. A medical institution has provided consultation on advance care planning to the declarant and affixed its seal on his or her advance decision.

2. The advance directive must be notarized by a notary public or witnessed by two or more persons with full disposing capacity.

3. The advance directive must be registered in his or her National Health Insurance IC card.

Therefore, formal ACP can include making and documenting an AD such as a written living will, which outlines a person's decisions about acceptance or refusal, in full or in part of EoL-care. This includes life-sustaining treatments and/or artificial nutrition and hydration under the specific clinical conditions and designating a surrogate decision-maker who can make decisions for the person in case of loss of competency. The PRAA stipulates that patient autonomy in healthcare must be respected to safeguard a person's rights to a good death, and to promote a harmonious physician-patient-relationship [16].

At present, both HPCA and PRAA coexist in clinical practice in Taiwan. Nevertheless, a discussion about death has long been taboo in Eastern society. Unless absolutely necessary, the topic of death is rarely formally discussed among family members. To date, the process of ACP remains extremely difficult for older Asian adults. Although ACP and signing ADs are based on person-centered care and personal autonomy, Chinese society emphasizes decisions that are made as a family or group and EoL-care is profoundly influenced by family and group values [17–19].

Studies have shown FCGs usually do not have adequate knowledge about the preferences of PWCIs for EOL-care [7, 20]. Therefore, when FCGs need to make EOL-care decision for PWCIs, they often feel uncertainty, stress, and guilt [21]. Family-centered care treats the family as a whole unit and embraces not only the view of PWCIs, but also family members [22]. Family-centered interventions may be more suitable for PWCIs because they can increase communication between the patient and family [23]. Family members, as well as FCGs acting as a health care agent, play an important role in helping PWCIs make future EOL-care decisions. Therefore, participation in the process of ACP can help FCGs better understand the values and preferences of PWCIs, which will be more conducive to the implementation of EOL-care decisions in accordance with the expectations of PWCIs [5].

A recent study demonstrated that an intervention that increases knowledge and attitudes regarding EOL-care can reduce decision-making conflicts for persons with mild dementia and

their FCGs [23]. However, the effect of the intervention on increasing the willingness to formally participate in ACP, EOL-care discussions and decision-making has not been explored. Therefore, the aim of this study was to examine the effect of a family-centered ACP intervention for PWCIs and FCGs focused on familiarity with ACP and its components, intent of PWCIs to engage in ACP and support of FCGs, participation in personal discussions about EoL-care, and consistency between PWCIs and their FCGs for life-sustaining treatment decisions.

## 2. Materials and methods

### 2.1. Participants

The study was conducted from March 2019 to September 2020. Dyads of PWCIs and their FCGs were recruited from outpatient clinics of regional teaching hospitals, dementia care centers, or community care centers located in Taipei City and New Taipei City, Taiwan. The inclusion criteria for PWCIs were: > 55 years of age; diagnosed with mild dementia or mild cognitive impairment based on a score of 0.5–1 on the Clinical Dementia Rating (CDR) scale; had never participated in an ACP consultation; and able to communicate in Mandarin or Taiwanese. Inclusion criteria for FCGs were: > 20 years of age and able to communicate in Mandarin or Taiwanese. Both PWCIs and their FCGs had to agree to participate in the study.

Sample size for this study was estimated for F tests with repeated measurements and analysis of variance, where $\alpha = 0.05$, power = 0.85, and effect size = 0.25 to 0.30 [24], which was estimated to be at least 33 dyads. However, attrition rate has been reported to be as high as 15% for relevant studies on ACP [21, 25]; therefore, we estimated 40 dyads were required.

A total of 114 PWCIs and FCGs met the inclusion criteria for participation in the intervention. However, 28 dyads were not included in the final analysis because the FCGs did not agree to participate (n = 16), contact information was incorrect (n = 6), or the intervention was not completed due to hospitalization of the patient (n = 2) or schedule conflicts (n = 4). Therefore, a total of 86 persons (44 dyads) completed the intervention; two caregivers cared for two PWCIs, thus they were repeatedly paired with those two participants for a total of 44 PWCIs and 42 FCGs (Fig 1).

### 2.2. Procedure

The study was conducted in accordance with the Declaration of Helsinki and approved by the Institutional Review Board (IRB) of Taipei City Hospital (TCHIRB-10605118-E) and the IRB of Cathay General Hospital (CGH-P 108066). PWCIs and their FCGs were informed of the study by flyers posted in the clinics and referrals from nurses, clinicians, and leaders of dementia support groups. The researchers explained the design and purpose of the study to all interested PWCIs and their FCGs; they were also assured of anonymity of the data. All participants provided written informed consent prior to the pre-intervention; PWCIs provided consent for collection of medical data from their charts. The IRBs in Taiwan require consent forms involving PWCIs that are easy-to-read, which requires a large-sized font and uses simple, easy-to-understand wording to describe the purpose and procedures of the study. We also invited FCGs to confirm that the PWCI understood the purpose of this study and was willing to participate. If any PWCI showed distress, appeared distracted, or provided an answer that was not relevant to the question, a reminder about the question or a short break was offered. If the response of the participant did not improve, the interview ceased. If either the PWCI or their FCG withdrew or terminated participation in the research, all data collection ceased for the dyads.

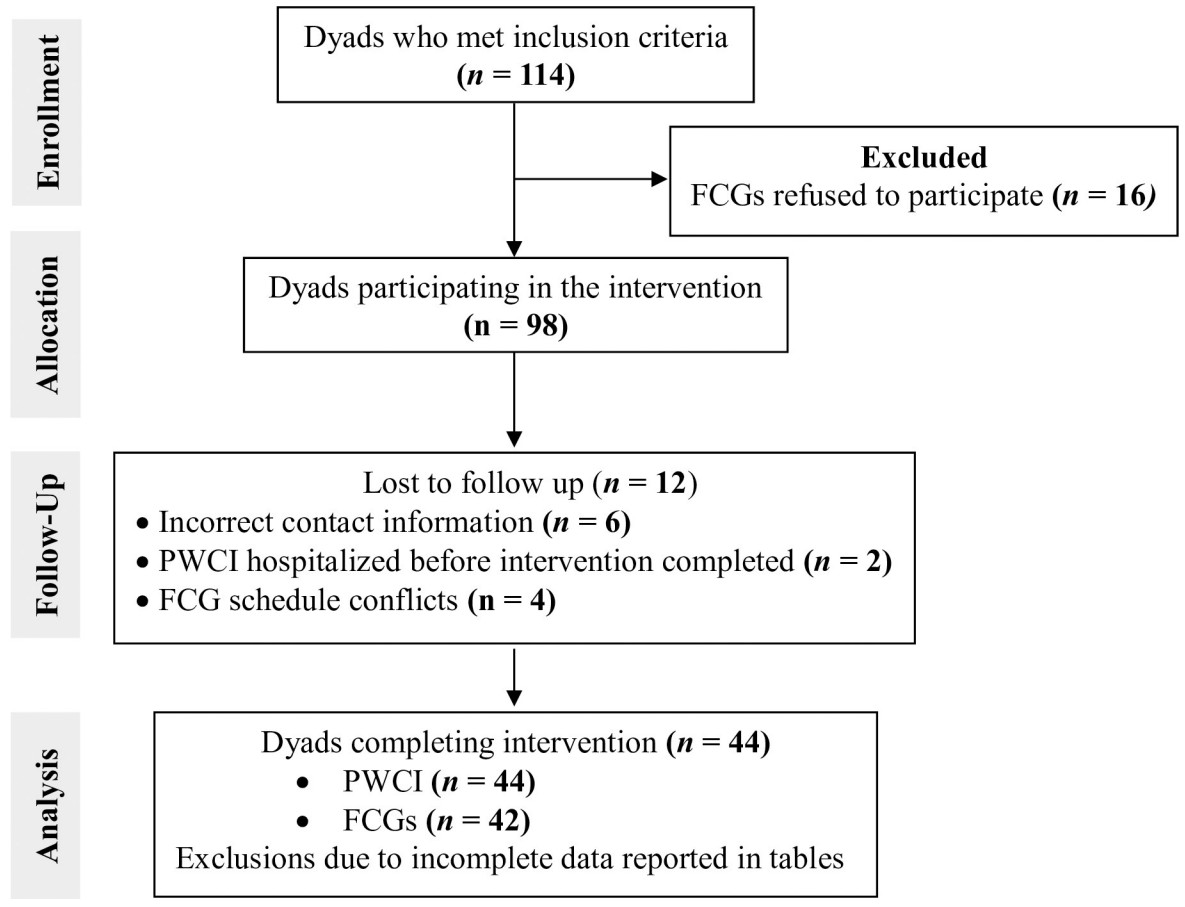

**Fig 1. CONSORT diagram for selection of dyads of family caregivers (FCGs) of persons with cognitive impairment (PWCI) participating in the ACP intervention.**

## 2.3. Family-centered ACP intervention

Prior to initiation of the intervention, pre-intervention data were collected from participants using four structured questionnaires, described below. The ACP information intervention was conducted in two parts by a senior nurse who was trained in explaining ACP to PWCIs and FCGs. In addition to PWCIs and FCGs, we invited other family members who were interested in an ACP information intervention, but these additional participants were not included in this study. The first part of the intervention provided an ACP manual to PWCIs and FCGs, titled, *Be My Own Master*: *Talk about Advance Care Planning for Dementia*. The manual included both text as well as pictorial aids, which have been shown to significantly improve understanding and decision-making for persons with mild to moderate dementia [26]. The nurse explained the contents of the manual, which included descriptions about the symptoms of end-stage dementia and the common EoL life-sustaining treatments, such as CPR, machine ventilation, tube feeding, intravenous infusion, and antibiotics. The manual also provided details about the benefits and risks of the treatments, as well as the process of establishing ACP and the regulations involved.

The second part of the intervention involved family-centered strategies for developing ACP and open discussions between PWCIs and their FCGs, which were facilitated by the nurse. Dyads were encouraged to communicate their thoughts and expectations about EoL care, and

to describe any uncertainties about the process; the nurse intervener provided any needed clarifications. The intervention lasted approximately 60 min. Four weeks after completing the intervention a posttest was conducted using the four structured questionnaires. Data collection was executed by a trained research assistance. Data for PWCIs and FCGs were collected separately.

## 2.4. Questionnaires

**2.4.1. Sociodemographic and clinical characteristics.** Structured questionnaires were used to collect data about demographic and clinical characteristics of the PWCIs, and demographic characteristics of the FCGs. Demographic data for all participants included age, gender, educational level and marital status. Clinical data for PWCIs included the presence of other diseases, months since diagnosis of dementia or mild cognitive impairment (MCI), awareness of their diagnosis of dementia or MCI, activities of daily living (ADLs) [27] and scores on the Cornell Scale for Depression in Dementia [28]. Demographic data for FCGs included religion, relationship to the PWCI, and the number of daily caregiving hours. Cognitive ability was determined with scores on the CDR and MMME, which were collected from participants' charts.

**2.4.2. Familiarity with ACP scale.** The 6-item familiarity with ACP scale was developed by the first author. Each item is a question pertaining to components of ACP for individuals in Taiwan: 1) the Patient Right to Autonomy Act, 2) ADs; 3) ACP consultations; 4) ADs for palliative care, 5) health care agents, and 6) DNR. Questions 1–3 were scored on a 3-point Likert scale: 0 = never heard about; 1 = heard about, content is unknown; 2 = heard about and know the content. Questions 4–6 were scored on a 4-point scale that included a fourth choice, 3 = know the content and signed an AD. The total scores ranged from 0–15 points; higher scores indicated more familiarity with the various components of ACP. In this study, the Cronbach's alphas for PWCIs and FCGs were 0.83 and 0.89, respectively.

**2.4.3. Intent to engage in ACP.** The intent to engage in ACP was measured with a researcher-developed 4-item scale for components of ACP: one scale was designed for PWCIs; a second scale was for FCGs. The scale for PWCIs involved items they intended to include in an ACP: 1) participation in a formal consultation about ACP; 2) AD for palliative care; 3) AD designating a healthcare agent; and 4) AD assigning their FCG as a proxy signer for a DNR. The scale for FCGS asked about items in the ACP they intended to support for the PWCI: 1) accompany the PWCI for a formal consultation about ACP; 2) AD for palliative care; 3) AD designating a health care agent; and 4) AD assigning the FCG as a proxy signer for a DNR. Each item was scored on a 5-point Likert scale from 1–5 (1 = definitely impossible or do not agree; 5 = definitely possible or completely agree). Total scores ranged from 4–20 points. For PWCIs, higher scores indicated a greater intention to engage in ACP; for FCGs, higher scores indicated a greater level of support for the intentions of the PWCI. The Cronbach's alphas for PWCIs and FCGs were 0.72 and 0.74, respectively.

**2.4.4. Personal discussions about EoL-care.** A 7-item scale for was developed to determine the number topics about EoL-care, unrelated to ACP, that had been discussed between PWCIs and FCGs. Items completed the question, "I have had private discussions with my family member about. . ." 1) using a proxy signer, 2) DNR, 3) use of a feeding tube, 4) palliative care, 5) placement in a nursing home facility, 6) dispersal of money and personal belongings, 7) and funeral/burial arrangements. Items were scored either No (have not discussed = 0) or Yes (have discussed = 1). Total scores ranged from 0–7; higher scores indicated a greater number of topics related to EoL-care that were discussed in private. The Cronbach's alphas for PWCIs and FCGs were 0.88 and 0.85, respectively.

**2.4.5. EoL-care decision scale.** The EoL-care decision scale was developed by the research team and was based on the literature for EoL treatments. This scale evaluated PWCIs and FCGs attitudes towards the use of three life-sustaining treatments for a person with severe dementia: CPR, an artificial respirator, or a feeding tube. Each item was scored on a 6-point Likert scale from (0) completely unacceptable to (5) completely acceptable. The Cronbach's alphas for PWCIs and FCGs were 0.93 and 0.94, respectively. The decision scores for each life-sustaining treatment were recoded and classified as either reject, accept or not sure. Therefore, a score = 0 was coded "1", indicating rejection of the treatment; an item score of 1 to 4, was coded "2", indicating not sure whether to accept or reject the life-sustaining treatment; and an item score = 5 was coded "3", indicating acceptance of the life-sustaining treatment. Consistency for each EoL-treatment decision between the PWCI and their FCG was assigned a value of 1 when there was agreement; otherwise, a value of 0 was assigned.

## 2.5. Statistical analysis

SPSS 22.0 software for Windows was used for statistical analysis. Descriptive statistics included frequency, percentage, mean, and standard deviation (SD). The effect of the intervention was determined with a paired sample t test by analyzing the difference in pre-intervention and post-intervention scores for familiarity with ACP and its components, intent to engage in ACP, and participation in informal discussions about the seven EoL-care treatments. Consistency between PWCIs and FCGs for decisions regarding EoL-care pre-intervention and post-intervention was determined with the Kappa parameter test [29]. The McNemar test was used to determine significant differences in scores for each topic of informal discussion and to compare consistency of EoL-care between decisions pre-intervention and post-intervention. A level of $p < .05$ was considered significant.

## 3. Results

Characteristics of PWCIs and FCGs are shown in Table 1. The mean age of PWCIs was 76.5 years; approximately one-third were widows; more than half were female and had a high school education or higher; most had been informed of or knew of their diagnosis of cognitive impairment or dementia (68%). The mean scores on the MMSE, CSDD, and ADL were 20.1, 4.4, and 93.3 respectively. One-third scored 0.5 on the CDR and 66% scored 1. The mean age of the FCGs was 57.1 years; 83% were female; 76% were married, and 83% had a high school education or higher. More than 60% FCGs were the child of the PWCI.

## 3.1. Effects of the ACP intervention on persons with cognitive impairment

**3.1.1. Effect of the intervention on familiarity with ACP.** Prior to participating in the ACP intervention, most PWCIs had never heard about the Patient Right to Autonomy Act (77.3%), ADs (75.0%), ACP consultations (77.3%), or AD for a health care agent (79.5%) and few knew about the content of these components (range = 2.3% - 11.4%). PWCIs were most familiar with the content for directives pertaining to DNR and palliative care (36.4% and 22.7%, respectively). One PWCI had signed an AD for palliative care, and another had signed an AD for a health care agent.

Post-intervention, 22.7% to 38.6% of PWCIs knew the content of these items, however few participants decided to sign any AD; only one additional PWCI signed an AD for palliative care and DNR (Table 2).

We also compared pre-intervention and post-intervention total scores and component scores for familiarity with ACP to determine if these changed post-intervention for PWCIs (Table 3). The total score for familiarity with ACP increased from 2.75 (SD = 3.40) to 5.31

**Table 1. Sociodemographic characteristics of persons with cognitive impairment (PWCIs) and family caregivers (FCGs).**

| Characteristics | PWCI (n = 44) | | FCGs (n = 42) | |
|---|---|---|---|---|
| | n (%) | Mean (SD) | n (%) | Mean (SD) |
| Age | | 76.5 (10.7) | | 57.1(11.8) |
| Gender | | | | |
| Male | 21 (47.7) | | 7 (16.7) | |
| Female | 23 (52.3) | | 35 (83.3) | |
| Marital status | | | | |
| Married | 32 (72.7) | | 32 (76.2) | |
| Single | | | 7 (16.7) | |
| Widowed or divorced | 12 (27.3) | | 3 (7.2) | |
| Educational level | | | | |
| ≤ Junior high school | 21 (47.7) | | 7 (16.6) | |
| ≥ High school | 23 (52.3) | | 35 (83.4) | |
| Persons with cognitive impairment | | | | |
| Other illnesses or diseases (range = 0–4) | | 1.6 (1.2) | | |
| None | 10 (22.7) | | | |
| 1 to 2 | 22 (50.0) | | | |
| 3 to 4 | 12 (27.3) | | | |
| Months since diagnosis (range = 1–108) | | 26.3 (25.6) | | |
| Informed or aware of diagnosis | | | | |
| Yes | 30 (68.2) | | | |
| No | 14 (31.8) | | | |
| MMSE score (range = 6–28) | | 20.1 (5.7) | | |
| CDR score (mild dementia range, 0.5–1.0) | | | | |
| 0.5 | 15 (34.1) | | | |
| 1.0 | 29 (65.9) | | | |
| CSDD (range = 0–24) | | 4.4 (6.1) | | |
| ADL (Range = 35–100) | | 93.3 (14.3) | | |
| Family caregivers only | | | | |
| Relationship to PWCI | | | | |
| Spouse | | | 14 (33.3) | |
| Child | | | 27 (64.3) | |
| Daughter-in-law | | | 1 (2.4) | |
| Hours of daily caregiving (range = 2-24h) | | | | 13.8 (9.3) |

Note: SD = standard deviation; MMSE = Mini-Mental State Examination; CDR = Clinical Dementia Rating scale; CSDD = Cornell scale for depression in dementia; ADL = activities of daily living; PWCI = persons with cognitive impairment

(SD = 5.21) post-intervention ($p < .001$). Except for DNR, all component scores increased significantly ($p = .01$ to $p < .001$).

**3.1.2. Effect of the intervention on ACP and personal discussions about EoL-care.** The effect of the family-centered ACP intervention on PWCIs was determined by the difference in mean scale scores pre-intervention and post-intervention (Table 3). The total score for intent to engage in ACP improved significantly post-intervention compared with pre-intervention scores (t = -2.52, $p = .016$). There was no change in the item score for engaging in a formal ACP discussion, however there was an increase in an AD for palliative care (t = -2.05, $p = .046$), a health care agent (t = -2.58, $p = .013$), and the FCG as a proxy signer for DNR (t = -2.55, $p = .015$).

**Table 2. Familiarity with components of ACP for PWCIs (n = 44), pre-intervention (pretest) and post-intervention (posttest).**

| | Never heard about | | Heard about, content unknown | | Heard about, know content | | Know content, signed AD | |
|---|---|---|---|---|---|---|---|---|
| | Pretest | Posttest | Pretest | Posttest | Pretest | Posttest | Pretest | Posttest |
| Component | n (%) | n (%) | n (%) | n (%) | n (%) | n (%) | n (%) | n (%) |
| PRAA | 34 (77.3) | 23 (52.3) | 9 (20.5) | 11 (25.0) | 1 (2.3) | 10 (22.7) | NA | NA |
| AD | 33 (75.0) | 18 (40.9) | 9(20.5) | 15 (34.1) | 2 (4.5) | 11 (25.0) | NA | NA |
| ACP Consultations | 34 (77.3) | 20 (45.5) | 8 (18.2) | 11 (25.0) | 2 (4.5) | 13 (29.5) | NA | NA |
| AD for palliative care | 29 (65.9) | 20 (45.5) | 4 (9.1) | 7 (15.9) | 10 (22.7) | 15 (34.1) | 1 (2.3) | 2 (4.5) |
| AD for healthcare agent | 35 (79.5) | 19 (43.2) | 3 (6.8) | 7 (15.9) | 5 (11.4) | 17 (38.6) | 1 (2.3) | 1 (2.3) |
| ADs for DNR | 18 (40.9) | 21 (47.7) | 10 (22.7) | 5 (11.4) | 16 (36.4) | 17 (38.6) | 0 (0.0) | 1 (2.3) |

Note: PWCIs = persons with cognitive impairment; PRAA = Patient Right to Autonomy Act; ADs = advance directives; ACP = advance care planning, NA = not applicable

**Table 3. Scale scores for PWCIs (n = 44): Familiarity with ACP, intent to engage in ACP, and participation in personal discussions about EoL-care and ACP pre-intervention (pretest) and post-intervention (posttest).**

| | Scores | | Paired t-test | | 95% CI | McNemar test |
|---|---|---|---|---|---|---|
| Scales/Items | Pretest | Posttest | t | p | lower, upper | p |
| **Familiarity with ACP**, M (SD) | | | | | | |
| Total score (range = 0–15) | 2.75 (3.40) | 5.31 (4.21) | -4.54 | <0.001 | -3.71, -1.43 | |
| Component scores | | | | | | |
| Patient Right to Autonomy Act (range = 0–3) | 0.25 (0.49) | 0.70 (0.82) | -3.81 | <0.001 | -0.70, -0.21 | |
| AD (range = 0–3) | 0.30 (0.55) | 0.84 (0.81) | -5.19 | <0.001 | -0.76, -0.33 | |
| ACP Consultation (range = 0–3) | 0.27 (0.54) | 0.84 (0.86) | -4.19 | <0.001 | -0.84, -0.30 | |
| AD for palliative care (range = 0–4) | 0.61 (0.92) | 0.98 (1.00) | -2.71 | 0.010 | -0.64, -0.09 | |
| AD for health care agent (range = 0–4) | 0.36 (0.78) | 1.00 (0.96) | -3.91 | <0.001 | -0.97, -0.31 | |
| AD for DNR (range = 0–4) | 0.95 (0.89) | 0.95 (0.99) | 0.00 | >0.999 | -0.32, 0.32 | |
| **Intent to engage in ACP**, M (SD) | | | | | | |
| Total score (range = 4–20) | 13.45(2.19) | 14.34 (2.63) | -2.52 | 0.016 | -1.60, -0.17 | |
| Item scores (range = 1–5) | | | | | | |
| Formal ACP consultation | 3.32 (0.93) | 3.45 (1.02) | -0.88 | 0.382 | -0.45, 0.18 | |
| AD for palliative care | 3.20 (1.03) | 3.52 (0.85) | -2.05 | 0.046 | -0.63, - 0.01 | |
| AD for a healthcare agent | 3.25 (0.89) | 3.64 (0.78) | -2.58 | 0.013 | -0.69, - 0.08 | |
| AD for FCG as proxy signer for DNR | 3.45 (0.85) | 3.77 (0.74) | -2.55 | 0.015 | -0.57, - 0.06 | |
| **Personal discussion about EoL-Care** | | | | | | |
| Total score, yes = 1 (range = 0–7), M (SD) | 1.04 (1.83) | 1.77 (2.30) | -2.42 | 0.020 | -1.33, -0.12 | |
| Personal discussions with FCGs (yes); n (%) | 16 (36.4) | 23 (52.3) | | | | 0.092 |
| Topics of discussions (yes); n (%) | | | | | | |
| Health care agent | 4 (9.1) | 10 (22.7) | | | | 0.109 |
| DNR | 11 (25.0) | 18 (40.9) | | | | 0.118 |
| Use of feeding tube | 9 (20.5) | 12 (27.3) | | | | 0.581 |
| Receiving palliative care | 6 (13.6) | 10 (22.7) | | | | 0.219 |
| Placement in a nursing home facility | 4 (9.1) | 7 (15.9) | | | | 0.453 |
| Dispersal of money, personal property | 3 (6.8) | 8 (18.2) | | | | 0.063 |
| Funeral and burial arrangements | 9 (20.5) | 13 (29.5) | | | | 0.344 |

Note: PWCIs = persons with cognitive impairment; M = mean; SD = standard deviation; CI = confidence interval; AD = advanced directive; EoL = End-of-Life; ACP = advanced care planning; DNR = do-not-resuscitate; DNR = do-not-resuscitate

The total score for participating in personal discussions about EoL-care also increased significantly post-intervention (t = -2.42, $p$ = .020). Although the personal discussions with FCGs increased from 36.4% to 53.2% post-intervention, the increase was not significant. There was also no significant increase in any of the seven item scores for specific topics of discussion.

## 3.2. Effects of the ACP intervention on family caregivers

### 3.2.1. Effect of the intervention on familiarity with ACP.

Changes in familiarity with ACP from pre- to post-intervention for FCGs are shown in Table 4. Pre-intervention, most FCGs had heard about many of the components of ACP but did not know about the specific contents. FCGs who knew the contents of ACP ranged from 26.2% to 45.2%. Similar to PWCIs, an AD for palliative care and DNR were the most familiar to FCGs (33.3% and 45.2%, respectively); before the intervention only 4.8% had signed a directive for a DNR.

Post-intervention, the contents of an AD related to palliative care and a DNR was understood by 71.4% and 69.0% of FCGs, respectively. Two additional FCGs signed an AD for palliative care; and one additional FCG signed an AD assigning a health care agent.

When we compared total scores and component scores for familiarity with ACP pre- and post-intervention for FCGs (Table 5), the total score increased significantly post-intervention (t = -7.89, $p < .001$). In addition, all six component scores for increased significantly post-intervention compared with pre-intervention ($p$ = .01 to $p < .001$).

### 3.2.2. Effects of the intervention on ACP and personal discussions about EoL-care.

The effect of the family-centered ACP intervention on FCGs was determined by the difference in mean scale scores pre-intervention and post-intervention (Table 5). On the questionnaire regarding intent to support the PWCI to engage in ACP, compared with pre-intervention scores, the total score increased post-intervention ($p < .05$) as did the item scores for AD for a health care agent and an AD for the FCG to act as a proxy signer for DNR (both $p < .05$). The total score on the questionnaire for participation in personal discussions about of EoL-care and ACP also increased post-intervention (t = −4.25, $p < .001$). Of the seven discussion topics, there was a significant increase in the percent of FCGs who had participated in personal discussions about a health care agent ($p < .05$), DNR ($p < .05$), use of a feeding tube ($p < .001$), and placement in a nursing home ($p < .05$).

## 3.3. Effects of the intervention on consistency for decisions about EoL-care between persons with cognitive impairment and family caregivers

Pre-intervention, only the consistency between PWCIs and FCGs for decisions about the use of CPR (65.9%) differed significantly (kappa $p$-value $< .05$). Post-intervention, percentages for

**Table 4. Familiarity with components of ACP for FCGs (n = 42), pre-intervention (pretest) and post-intervention (posttest).**

| | Never heard about | | Heard about, content unknown | | Heard about, know content | | Know content, signed AD | |
| | Pretest | Posttest | Pretest | Posttest | Pretest | Posttest | Pretest | Posttest |
| Component | n (%) | n (%) | n (%) | n (%) | n (%) | n (%) | n (%) | n (%) |
|---|---|---|---|---|---|---|---|---|
| PRAA | 6 (14.3) | 2 (4.8) | 25 (59.5) | 8 (19.0) | 11 (26.2) | 32 (76.2) | NA | NA |
| AD | 8 (19.0) | 1 (2.4) | 21(50.0) | 5 (11.9) | 13 (31.0) | 36 (85.7) | NA | NA |
| ACP Consultations | 8 (19.0) | 1 (2.4) | 22 (52.4) | 5 (11.9) | 12 (28.6) | 36 (85.7) | NA | NA |
| AD for palliative care | 7 (16.7) | 1 (2.4) | 20 (47.6) | 10 (23.8) | 14 (33.3) | 29 (69.0) | 0 (0.0) | 2 (4.8) |
| AD for healthcare agent | 15 (35.7) | 4 (9.5) | 16 (38.1) | 7 (16.7) | 11 (26.2) | 30 (71.4) | 0 (0.0) | 1 (2.4) |
| ADs for DNR | 8 (19.0) | 2 (4.8) | 13 (31.0) | 8 (19.0) | 19 (45.2) | 30 (71.4) | 2 (4.8) | 2 (4.8) |

Note: FCGs = family caregivers; PRAA = Patient Right to Autonomy Act; ADs = advance directives; ACP = advance care planning, NA = not applicable

**Table 5. Scale scores for FCGs (n = 44): Familiarity with ACP, intent to engage in ACP, and participation in personal discussions about EoL-care and ACP pre-intervention (pretest) and post-intervention (posttest).**

| | Scores | | Paired t-test | | 95% CI | McNemar test |
|---|---|---|---|---|---|---|
| Scales/Items | Pretest | Posttest | t | p | lower, upper | p |
| **Familiarity with ACP[1], Mean (SD)** | | | | | | |
| Total score (range = 0–15) | 6.81 (3.65) | 10.57 (2.50) | -7.89 | <0.001 | -4.73, -2.80 | |
| Component scores | | | | | | |
| Patient Right to Autonomy Act (range = 0–3) | 1.11 (0.62) | 1.72 (0.54) | -6.59 | <0.001 | -0.80, -0.43 | |
| AD (range = 0–3) | 1.14 (0.70) | 1.84 (0.43) | -6.66 | <0.001 | -0.92, -0.49 | |
| ACP Consultation (range = 0–3) | 1.13 (0.68) | 1.84 (0.43) | -6.64 | <0.001 | -0.95, -0.51 | |
| AD for palliative care (range = 0–4) | 1.23 (0.74) | 1.77 (0.57) | -4.96 | <0.001 | -0.77, -0.32 | |
| AD for health care agent (range = 0–4) | 0.91 (0.77) | 1.68 (0.67) | -5.97 | <0.001 | -1.03, -0.51 | |
| AD for DNR (range = 0–4) | 1.36 (0.84) | 1.77 (0.60) | -3.74 | 0.001 | -0.63, -0.19 | |
| **Intent to support PWCI to engage in ACP[2]** | | | | | | |
| Total score (range = 4–20), M (SD) | 15.29 (1.81) | 15.98 (2.14) | -2.56 | 0.014 | -1.22, -0.14 | |
| Item scores (range = 1–5), M SD | | | | | | |
| Accompany PWCI for formal ACP consultation | 3.88 (0.74) | 3.93 (0.81) | -0.35 | 0.728 | -0.32, 0.23 | |
| AD for palliative care | 3.77 (0.87) | 3.84 (0.72) | -0.50 | 0.618 | -0.35, 0.21 | |
| AD for a healthcare agent | 3.57 (0.70) | 3.93 (0.73) | -3.09 | 0.020 | -0.61, -0.04 | |
| AD for FCG as proxy signer for DNR | 4.07 (0.62) | 4.27 (0.66) | -2.15 | 0.037 | -0.40, -0.01 | |
| **Personal discussion about EoL-Care[2]** | | | | | | |
| Total score, yes = 1 (range = 0–7); M (SD) | 1.66 (1.89) | 2.89 (2.47) | -4.25 | <0.001 | -1.81, -0.64 | |
| Personal discussions with FCGs (yes); n (%) | 33 (75.0) | 41 (93.2) | | | | 0.008 |
| Topics of informal discussion (yes); n (%) | | | | | | |
| Health care agent | 5 (11.4) | 18 (40.9) | | | | 0.002 |
| DNR | 16 (36.4) | 26 (59.1) | | | | 0.013 |
| Use of feeding tube | 10 (22.7) | 22 (50.0) | | | | <0.001 |
| Receiving palliative care | 17 (38.6) | 19 (43.2) | | | | 0.774 |
| Placement in a nursing home facility | 5 (11.4) | 13 (29.5) | | | | 0.039 |
| Dispersal of money, personal property | 4 (9.1) | 9 (20.5) | | | | 0.063 |
| Funeral and burial arrangements | 16 (36.4) | 20 (45.5) | | | | 0.344 |

Note: PWCIs = persons with cognitive impairment; M = mean; SD = standard deviation; CI = confidence interval; AD = advanced directive; EoL = End-of-Life; ACP = advanced care planning; DNR = do-not-resuscitate; DNR = do-not-resuscitate

[1] n = 42

[2] n = 44

consistency increased (ranging from 72.7–77.3%) as well as kappa values (ranging from 0.31–0.37). Although kappa *p*-values indicated significant increases post-intervention in consistency between PWCIs and FCGs for CPR, artificial respirator, and feeding tube between increased (all *p* < .05), the McNemar test indicated there was no significant difference in consistency between pre- and post-intervention scores (Table 6).

## 3.4. Comparison of the effect of the ACP intervention for PWCIs with different clinical dementia ratings

**3.4.1. Scale scores for PWCIs with CDR scores of 0.5 and 1.0.** We examined if the effect of the intervention differed between PWCIs with different scores on the CDR scale (Table 7).

**Table 6. Consistency between PWCIs and FCGs for life-sustaining End-of-Life (EoL) decisions pre- and post-intervention (n = 44 dyads).**

| | Pre-intervention | | | | | Post-intervention | | | | | McNemar |
| | | | | Kappa components | | | | | Kappa components | | test |
| | PWCIs | FCGs | Consistency | K | p | PWCIs | FCGs | Consistency | K | p | p |
| EoL decision[a] | n (%) | n (%) | n (%) | | | n (%) | n (%) | n (%) | | | |
|---|---|---|---|---|---|---|---|---|---|---|---|
| CPR | 26 (59.1) | 29 (65.9) | 29 (65.9) | 0.34 | 0.006 | 33 (75.0) | 35 (79.5) | 34 (77.3) | 0.37 | 0.006 | 0.267 |
| Respirator | 29 (65.9) | 28 (63.6) | 28 (63.6) | 0.25 | 0.059 | 33 (75.0) | 34 (77.3) | 33 (75.0) | 0.32 | 0.024 | 0.359 |
| Feeding tube | 29 (65.9) | 27 (61.4) | 28 (63.6) | 0.22 | 0.092 | 32 (72.7) | 33 (75.0) | 32 (72.7) | 0.31 | 0.024 | 0.481 |

Note: PWCI = persons with cognitive impairment; FCGs = family caregivers; CPR = cardiopulmonary resuscitation

[a] Reject = 1

Pre-intervention, only familiarity with ACP differed significantly between PWCIs with a CDR = 0.5 and 1 (t = 2.12, $p < .05$). Post-intervention, there was no significant difference between the two groups of PWCIs any of the components of ACP (Table 7).

**3.4.2. Scale scores for FCGs of PWCIs with CDR scores of 0.5 and 1.0.** We also examined whether component scores differed between FCGs of PWCIs based on dementia ratings pre- and post-intervention. There was no significant difference for any variable between FCGs of PWCIs with a CDR score of 0.5 or 1.0 pre-intervention or post-intervention (Table 8).

# 4. Discussion

This study is the first to explore the effects of a family-centered ACP intervention on familiarity with ACP and its components, intent to engage in ACP, and participation in personal discussions about EoL-care for both PWCIs and their FCGs. After completion of the intervention there was a significant improvement in familiarity with components of ACP, total score for intent to engage in ACP for PWCIs and support from FCGs, and participation in informal discussions about topics related to EoL-care discussions. There were only modest increases in consistency between PWCIs and the FCGs for life-sustaining treatment decisions. The following sections discuss the effectiveness of the intervention on familiarity with ACP, intent to

**Table 7. Comparison of pretest/posttest scale scores between PWCIs with different CDRs (n = 44).**

| | Pretest | | | Posttest | | |
| | CDR score | | | CDR score | | |
| Scale/item scores | 0.5 | 1.0 | t/$\chi^2$ | 0.5 | 1.0 | t/$\chi^2$ |
|---|---|---|---|---|---|---|
| Familiarity with ACP, M (SD) | 4.20 (3.57) | 2.00 (3.11) | 2.12* | 6.00 (4.12) | 4.97 (4.28) | 0.77 |
| Intent to engage in ACP, M (SD) | 13.00 (2.24) | 13.69 (2.17) | -0.99 | 13.87 (2.97) | 14.59 (2.46) | -0.86 |
| Personal EoL-Care discussions, M(SD) | 0.60 (1.80) | 1.28 (1.83) | -1.17 | 1.27 (2.05) | 2.03 (2.41) | -1.05 |
| EoL-Care decision[a], n (%) | | | | | | |
| CPR | 10 (66.7) | 16 (55.2) | 2.92 | 11 (73.3) | 22 (75.9) | 0.68 |
| Respirator | 11 (73.3) | 18 (62.1) | 1.29 | 10 (66.7) | 23 (79.3) | 0.84 |
| Feeding tube | 10 (66.7) | 19 (65.5) | 1.86 | 10 (66.7) | 22 (75.9) | 2.07 |
| Consistency with FCGs | | | | | | |
| CPR | 11 (73.3) | 18 (62.1) | 0.56 | 13 (86.7) | 21 (72.4) | 1.14 |
| Use of respirator | 10 (66.7) | 18 (62.1) | 0.09 | 12 (80.0) | 21 (72.4) | 0.30 |
| Use of feeding tube | 10 (66.7) | 18 (62.1) | 0.09 | 12 (80.0) | 20 (69.0) | 0.61 |

Note: PWCIs = persons with cognitive impairment; CDR = Clinical Dementia Rating scale; ACP = advanced care planning; M = mean; SD = standard deviation; FCGs = family caregivers; EoL = End-of-Life; DNR = do-not-resuscitate; CPR = cardiopulmonary resuscitation

[a] Reject = 1

* $p < .05$

**Table 8. Comparisons of pretest/posttest scale scores between FCGs of PWCIs with different CDRs.**

| Scale/item scores | Pretest | | | Posttest | | |
| --- | --- | --- | --- | --- | --- | --- |
| | CDR score | | | CDR score | | |
| | 0.5 | 1.0 | t/$\chi^2$ | 0.5 | 1.0 | t/$\chi^2$ |
| Familiarity with ACP[1], M (SD) | 7.87 (3.23) | 6.34 (3.73) | 1.34 | 11.00(1.73) | 10.45(2.78) | 0.70 |
| Intent to engage in ACP[2], M (SD) | 15.40 (2.06) | 15.37 (1.80) | 0.05 | 15.57 (1.83) | 16.03 (2.33) | -0.65 |
| Personal EoL-Care discussions, M(SD) | 1.47 (1.88) | 1.76 (1.92) | -0.48 | 2.73 (2.37) | 2.97 (2.56) | -0.29 |
| EoL-Care decision[a], n[2] (%) | | | | | | |
| CPR | 10 (66.7) | 19 (65.5) | 0.54 | 13 (86.7) | 22 (75.9) | 0.96 |
| Respirator | 9 (60.0) | 19 (65.5) | 0.29 | 13 (86.7) | 21 (72.4) | 1.34 |
| Feeding tube | 10 (66.7) | 19 (65.5) | 1.86 | 10 (66.7) | 22 (75.9) | 2.07 |

Note: FCGs = family caregivers; PWCIs = persons with cognitive impairment; CDR = Clinical Dementia Rating scale; ACP = advanced care planning; M = mean; SD = standard deviation; EoL = End-of-Life; CPR = cardiopulmonary resuscitation

[a] Reject = 1

[1] n = 42

[2] n = 44

engage in ACP, participation in personal discussions about EoL-care, and consistency of decisions among PWCI and their FCGs.

## Effects of intervention on familiarity with ACP

Pre-intervention scores demonstrated most PWCIs and FCGs were unfamiliar with ACP-related information. Nearly 80% of PWCIs had never heard about it; and those who had heard the various terms did not know the content. The lack of familiarity and awareness of ACP directives are the main barriers for individuals to engage in ACP [30]. This is supported by the significant increase in post-intervention scores for familiarity with ACP for both PWCIs and FCGs and is consistent with prior studies [14, 25, 31].

Only one PWCI had heard about and understood the contents of the PRAA pre-intervention and post-intervention the number of participants remained low (n = 10; 22.7%). Although only 26% of FCGs (n = 11) had heard about and understood the contents of the PRAA pre-intervention, post-intervention the number rose to 32 (76.2%). These findings might be related to the more recent implementation of the PRAA in 2019. The participants with mild dementia or cognitive impairment may not have had the mental capacity to absorb the information about the PRAA, even with the intervention. Whereas the PWCI had previous knowledge of the HPCA, which has been in effect for more than 20 years. Familiarity with ADs and the content of ADs and ACP were also low for PWCIs. In contrast, FCGs familiarity with and content of ADs, ACP, and PRAA pre- and post-intervention were similar. Awareness about the HPCA has been shown to be better for aspects of the HPCA, such as AD for palliative care or DNR for FCGs compared with PWCIs [2]. Familiarity with ADs for assigning a health care agent was low for all participants pre-intervention. This might be explained by the fact that in Chinese cultures family decision-making is highly valued and healthcare professionals rarely promote the idea of assigning a designated health care agent when discussing ACP. Therefore, PWCIs and FCGs are less aware of this option for an AD.

## Effects of intervention on intent to engage in ACP

It is worth noting that there was no difference in pre- and post-intervention scores for PWCIs or FCGs to participate in a formal ACP consultation. The lack of statistical significance may be

due to gaps between practice and policy. Providing individuals with a formal ACP consultation in Taiwan is a recent concept, as it is a new provision at the core of the PRAA [16]. The purposes of an ACP consultation are to ensure the patient, family members, and medical service providers have sufficient information to make informed decisions regarding preferences for life-sustaining EoL medical treatments before the patient signs any AD. At present, only designated medical institutions in Taiwan have ACP consultation clinics and low availability and accessibility can reduce an individual's willingness to participate in consultations [30]. Policy-makers should examine ways to increase the accessibility and reduce obstacles to participating in consultations about ACP, especially for PWCIs and their FCGs.

Post-intervention, there was a significant increase in PWCIs' scores for an AD for a health care agent and an AD for a FCG as proxy signer for DNR. FCGs' scores supporting these ADs also increased significantly. Previous studies have reported that PWCIs are less likely to participate in or sign an AD on their own initiative [17, 31] or, due to fear of making the wrong decisions, may delegate decisions and authority to family members [31]. Therefore, our findings suggest the opportunities for dialogue and communication provided by the family-centered ACP intervention may have allowed PWCIs and FCGs to increase mutual understandings about preferences for ADs, including the possibility of designating a health care agent [3].

Post-intervention scores increased modestly for PWCIs intent to have an AD for palliative care, however this item score did not increase for FCGs. These findings are similar to an integrative literature review on the impact of ACP interventions for PWCIs and caregivers, which found family members could act as facilitators or barriers to ACP for PWCIs [32]. A study by Lewis et al. (2015) introduced ACP to persons with mild cognitive impairment or newly diagnosed dementia and their FCGs who were enrolled in a memory clinic [14]. Their results showed that 4% of FCGs and 9% of PWCIs completed AD documentation. Lo et al. (2017) provided 158 PWCIs with a psycho-educational intervention about ACP and 10.6% of participants chose to complete an AD after the 12-month intervention [31]. We speculate that the shorter 4-week follow-up period in our study may explain the lack of why only a few participants completed an AD post-intervention. Although recent studies suggest ADs are often triggered by a friend or family member experiencing health changes or a traumatic life event [33] and some individuals prefer informal plans over written AD documentation [30]. We did not collect information about changes in family members' health status or preferences in AD documentation. Therefore, it will be important to include this information in future ACP intervention studies.

ACP readiness can vary according to need and participants may have felt that their current plans and discussions with family were sufficient [14, 34, 35]. ACP is a process involving social interactions, and the effect of an ACP intervention that is only provided once may be limited. Past studies suggest that ACP in dementia care should be started as soon as the diagnosis is made and then revisited with both PWCIs and their family members on a regular basis and following any significant change in health condition [1, 23]. Our intervention is the first step in drawing attention to the autonomy of life and to initiate a dialogue between PWCIs and FCGs. Incorporating ACP into clinical care for PWCIs with a gradual increase in information and supporting discussions between PWCIs and FCGs may increase completion of ADs [14, 23].

## Effect of the intervention on EoL-care discussions

Only FCGs had a significant increase in participation in personal discussions about EoL-care post-intervention. These findings are in partial agreement with a meta-analysis of randomized controlled trials on the efficacy of ACP interventions, which demonstrated an increase in dialogue about EoL-Care decision-making between adults and family members [36]. Issues

surrounding death, such as EoL-care decisions, are considered taboo in Chinese society, including Taiwan [2, 37]. Our intervention created an opportunity to initiate dialogue and stimulate more discussion on issues related to EoL between PWCIs and their FCGs through professional guidance. However, in terms of the number of topics of discussion, there was a significant increase for FCGs, which was for a health care agent, DNR, use of a feeding tube, and placement in a nursing home. Only 36% of PWCIs had discussed EoL decisions with their family. This result supports the finding of a study by Hirschman et al. (2008) who reported PWCIs were less likely to actively discuss EoL-care matters compared with FCGs [33].

Our ACP intervention included information about the symptoms of advanced dementia, such as eating problems and a decline in capacity for decision-making. This may explain the significant increase in post-intervention personal discussions for FCGs on the topics of feeding tubes and health care agents. Both PWCIs and FCGs were unfamiliar with using an AD to assign a health care agent prior to the intervention. However, post-intervention discussions with PWCIs about an AD for a health care agent was one of the topics that increased significantly for FCGs. One explanation for this change is that the intervention may have increased FCGs awareness that they might not have sufficient authority to make EoL–care decisions for their relative with cognitive impairment [17, 38]. One study reported FCGs believed that if a PWCI designated a surrogate as early as possible, the uncertainty and conflict of decision-making at EoL might be reduced [39].

### Effects of the intervention on consistency of life-sustaining treatment decisions

Although the post-intervention scores increased for the decision to reject life-sustaining EoL medical treatments and consistency for life-sustaining treatment decisions for both FCGs and PWCIs, they were not significantly different compared with pre-intervention scores. These findings are similar to recent studies demonstrating PWCIs are more likely to prefer life-sustaining treatments compared with persons without cognitive decline and their FCGs [20, 40]. Other studies have reported FCGs to have a low to moderate consistency with PWCIs on preferences for life-sustaining treatment [18, 20, 41]. This may be due to the inability of FCGs to clearly understand the preferences of the PWCI, even though the topic had been discussed. This disconnect is supported by several studies that have reported discussing preferences for EoL treatments with a family member or with a designated health care agent does not guarantee predictive accuracy [18, 20, 42].

The decrease in a desire for the use of life-sustaining treatments at EoL might be explained by limitations due to cognitive impairment, particularly changes related to short-term memory loss, which may make it difficult for PWCIs to understand the consequences of their preferences in a hypothetical situation [40, 43]. Perceptions may also differ between PWCIs and FCGs regarding their discussions about life-sustaining treatments [18] and the effectiveness of ACP may depend on family members' understanding of patient's preferences [44].

Earlier studies found milder impairment of cognitive function was associated with a greater likelihood of engagement in ACP [45, 46]. However, our study found the only difference between PWCI s with a CDR score of 0.5 and 1.0 was for pre-intervention scores for familiarity with ACP. There were no significant differences for pre-intervention scores for FCGs of the two groups of PWCIs or post-intervention scores for PWCIs or FCGs.

The design of the ACP intervention was guided by family-centered care, which included use of graphic manuals, simple text, and easy-to-understand terms to help both PWCIs and their FCGs in understanding common life-sustaining medical treatments, and the procedures involved in formal ACP. The educational level of the PWCIs was lower than the FCGs, which

could explain the lower scores for PWCIs compared with FCGs if they were less able to understand and absorb the contents of the intervention. PWCIs were secure in expressing their personal thoughts and feelings regarding EoL-care due to the intimacy of their relationship with their FCGs, which lead to the acquisition of new knowledge between the dyads, despite the differences in dementia ratings. The family-centered intervention assisted PWCIs to participate in ACP and express their wishes about EoL-care to FCGs. Family-centered ACP should be conducted and promoted before the decision-making capacity of PWCIs is compromised, which will ensure that preferences are respected and fulfilled.

### Research limitations

This study had several limitations, which might prevent generalization of our findings. First, a single-group paired with pretest-posttest design was used and data were collected from participants recruited from only regional hospitals in northern Taiwan and this Metro Area of Taipei has a population with high levels of education. Thus, our findings might not be relative to other areas or with lower levels of education. Second, this study required pairing PWCIs and FCGs, which increased the difficulty of obtaining a larger sample size. Third, because 16 FCGs refused to participate in the study (14%), the possibility that the acceptance level for EoL issues was overestimated should be a concern. The issues of EoL and death remain culturally sensitive topics in Taiwan and reluctance to discuss EoL might have been the reason 16 FCGs refused to participate. Therefore, we suggest adding questions about why FCGs refused participation or withdrew at each stage, which could benefit the future design and relevance of the research findings. Fourth, hypothetical life-sustaining treatment decisions and ACP intentions may not match real-life situations or the educational level of participants. We recommend the model of the ACP intervention and the material provided be further tested to determine whether these are suitable for people with different backgrounds and levels of education. Finally, although the ACP intervention increased the number of topics for EoL-care discussions, the details of the discussions cannot be known, and qualitative supplementation is recommended in the future.

## 5. Conclusions

To the best of our knowledge, this study is the first to examine the effects of a family-centered ACP intervention model in an Asian setting. Our findings demonstrated that a family-centered ACP intervention can create opportunities for PWCIs to participate in decision-making about their EoL-care and increase EoL-care discussions between PWCIs and their FCGs. However, post-intervention there was no increase for participants in their intention to participate in a formal ACP consultation or to complete forms for AD. We suggest that a family-centered ACP intervention should be incorporated into routine clinical care in conjunction with encouraging ongoing communication between PWCIs and their FCGs, which might increase completion of ACP and ensure preferences for EoL-care are respected for PWCIs.

### Supporting information

**S1 File.**
(XLS)

### Acknowledgments

We would like to thank all the PWCIs and FCGs who took part in the study. We also thank all healthcare professionals to refer participants for this study.

## Author Contributions

**Conceptualization:** Hsiu-Li Huang, Wei-Ru Lu, Huei-Ling Huang, Chien-Liang Liu.

**Data curation:** Hsiu-Li Huang, Huei-Ling Huang.

**Formal analysis:** Hsiu-Li Huang, Wei-Ru Lu.

**Funding acquisition:** Hsiu-Li Huang.

**Investigation:** Wei-Ru Lu.

**Methodology:** Hsiu-Li Huang, Wei-Ru Lu, Huei-Ling Huang, Chien-Liang Liu.

**Project administration:** Hsiu-Li Huang.

**Resources:** Chien-Liang Liu.

**Supervision:** Hsiu-Li Huang, Chien-Liang Liu.

**Validation:** Hsiu-Li Huang, Huei-Ling Huang, Chien-Liang Liu.

**Visualization:** Hsiu-Li Huang.

**Writing – original draft:** Hsiu-Li Huang.

**Writing – review & editing:** Hsiu-Li Huang.

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
