## [Decision Letter · Decision Letter 0]

29 Jun 2021

PONE-D-21-14530

The effect of a family-centered advance care planning intervention for persons with cognitive impairment and their family caregivers on end-of-life care discussions and decisions

PLOS ONE

Dear Dr. Huang,

Thank you for submitting your manuscript to PLOS ONE. After careful consideration, we feel that it has merit but does not fully meet PLOS ONE’s publication criteria as it currently stands. Therefore, we invite you to submit a revised version of the manuscript that addresses the points raised during the review process.

A rebuttal letter that responds to each point raised by the academic editor and reviewer(s). You should upload this letter as a separate file labeled 'Response to Reviewers'.A marked-up copy of your manuscript that highlights changes made to the original version. You should upload this as a separate file labeled 'Revised Manuscript with Track Changes'.An unmarked version of your revised paper without tracked changes. You should upload this as a separate file labeled 'Manuscript'

We look forward to receiving your revised manuscript.

Kind regards,

Helen Chan

Academic Editor

PLOS ONE

Additional Editor Comments (if provided):

The study is timely and meaningful, but further elaboration to improve its clarity is needed.

Reviewers' comments:

Reviewer's Responses to Questions

**Comments to the Author**

1. Is the manuscript technically sound, and do the data support the conclusions?

Reviewer #1: Partly

Reviewer #2: Yes

2. Has the statistical analysis been performed appropriately and rigorously? 

Reviewer #1: Yes

Reviewer #2: Yes

3. Have the authors made all data underlying the findings in their manuscript fully available?

Reviewer #1: Yes

Reviewer #2: Yes

4. Is the manuscript presented in an intelligible fashion and written in standard English?

Reviewer #1: No

Reviewer #2: Yes

5. Review Comments to the Author

Reviewer #1: It’s my pleasure to review your manuscript. This paper presents a very important research topic—using a family-centered approach to promote ACP to persons with cognitive impairment (PWCIs) and their family caregivers (FCGs) which has drawn much attention in the research world and clinical practice. The aim of this paper was to evaluate the effectiveness of the ACP intervention on four outcomes: i) Familiarity with ACP; ii) ACP engagement intention; iii) participation in EOL care discussion; iv) dyadic concordance on EOL care decisions between the PWCIs and FCGs.

Methodology section

This section was clear in general. Description of study design and intervention were clear. The eligibility criteria was somewhat clear. Mind the way of using “or” to link up different conditions. It was somehow misleading that individuals with MMSE≥18 was eligible to participate. Also, in the result section, it was mentioned that two PWCIs had signed AD at baseline. However, according to the eligibility criteria, individuals who have participated in ACP seemed to be excluded from this study. Would it be the case in Taiwan if an individual can sign AD without went through ACP? If not, screening of subjects may have some problems. A CONSORT diagram is recommended to summarize the subject recruitment picture. “PWCIs” and “persons with dementia” were used interchangeably that may confuse the readers.

This study employed self-developed questionnaires as outcome measures. The items of each questionnaire were clearly presented, yet some of the contents and scoring method was unclear. For the 7-item scale for informal discussions about EOL care, the meaning of “informal discussion” was unclear. Please clarify if the ACP intervention itself a formal or informal discussion under your operational definition. To measure the concordance on EOL care between the dyads, a hypothetical scenario was used. The ratings of the Likert scale from “0” to “5” were unclear. Would there be an option representing “not sure”? Such response was absolutely relevant and should be included. Please elaborate if the recoding method would have altered the original response.

Result section

Mind the way of presentation. Some of the results were not aligned with the sub-headings. For example, the t-test results of Familiarity of ACP should be put under 3.1.1, instead of 3.1.2. Same problem appeared in 3.2.1 from the FCGs’ perspective.

Under 3.3, the interpretation of “consistency was significantly different between PWCIs and FCGs” were not supported by data.

Discussion section

Two references were used to compare with the results of this study on intention to engage in ACP. However, these two studies measured AD completion as outcome, which was far different from yours. Also, Lewis et al. (2015) reported that their recruitment period spanned through 8 months. The authors misinterpreted that their outcomes were measured at 8 months after the intervention. The conclusion on limited effectiveness for once-off ACP intervention to increase AD completion was not supported by the results.

Regarding 4.3, “effect of the intervention on EOL care discussion, the topic sentence of this paragraph was not supported by results. Participation in EOL discussion for PWCIs did not show significant difference (p= 0.092). Some discussions were made to explain the descriptive statistics of pre-test results, which was not aligned with the sub-heading.

Overall, this paper provided very valuable data on the effectiveness of an ACP programme. The manuscript could be presented in a more logical way to enhance readers’ understanding. Also, the manuscript requires professional editing before it can be published.

Reviewer #2: First of all, the study was worthwhile to explore and provided a good intention to understand the advance care planning in Taiwan. Moreover, the data collection and results did support the findings to reveal particular situations of selected participants. Indeed, there are several points for the author's consideration and it may need for the author to improve for a better explanation of this topic:

(1) The understanding between advance care planning and advance directive should be elaborated further. As the author focused much on medical perspectives and items in this research, it might need to let the readers have an explicit notification about the concept of advanced care planning, including social and medical aspects for a holistic care consideration. It might be good to clarify the ideas of advance care planning in the parts of discussion or limitation in this study;

(2) One key element of this research was about "family-centered" approach and it will be necessary to explain this concept in this article. "Family-centered" intervention was one of the significant ideas to drive this study out and strongly advised the author to define or explain more in this aspect. The writing did not cover much or even limited discussion for this vital concept. The literature reviews or the discussion of findings were suggested to integrate understanding "family-centred" intervention. It would fulfil the reasonable expectation of readers and echo to author's topic with a particular "family centered" advance care planning intervention.

(3) The difference between people with mild cognitive impairment and people with dementia should make a warm reminder to the readers. As CDR 0.5 and CDR 1 included one key consideration of differences which was the status of "reversible" and "irreversible". A person with CDR 0.5 might have around 10-15% possibility of becoming dementia per year if there was no active intervention. The condition and context of this group of participants will not be similar to those who were at CDR 1 (diagnosis of dementia which was "irreversible"). Having a bit of clarification or even interest in exploring any difference between these two groups of participants. Their family caregivers might also have different interpretations and willingness of discussing advance care planning. As the study contained 34.1% (CDR=0.5) and 65.9% (CDR=1) of participants, the author might find out further in the results and findings of this research;

(4) For the consent form, the author might consider designing a unique consent form for people with dementia. In regard to the ethical consideration and a good standard of research involving people with dementia, a bigger size of wordings with simple terms and cue cards for people with dementia to understand the research might be offered. It might be an option for the author to think over in the future development of study relating to people with dementia;

(5) For the discussion part, the author mentioned the consistency between PWCIs and FCGs for life-sustaining treatment decisions. It would be great if more reviews could be included for understanding this discrepancy. The educational levels of the participants in this research could be considered and how "family centered" interventions could improve this situation would be valuable discussion in this manuscript. The author identified this remarkable area to explore and it would be excellent if the author could provide some of the reflections as well as recommendations in this part;

(6) Minor typo at row 430;

(7) For the limitation part, it would be good to report 16 FCGs who refused to participate in this study. If possible, the questionnaires could add some questions to understand the reasons which might help the author explore further in this topic. It will be fascinating to explore the reason for withdrawal in each stage to benefit the future design and relevant research findings. It also contributes to the accumulation of knowledge.

(8) The author might consider suggesting the optimal time to discuss advance care planning in dementia care, i.e. during the post-diagnostic period or whenever the health condition was changed, etc. The conclusion might be supported by previous literature for echoing the critical ideas of "family centered" interventions in this study.

Thanks for conducting this study which contributed to the development of advance care planning for people with cognitive impairments. That would be a significant area to explore more understand and shared practical wisdom in the coming time. Good luck and best wishes~~

6. PLOS authors have the option to publish the peer review history of their article (what does this mean?). If published, this will include your full peer review and any attached files.

Reviewer #1: No

Reviewer #2: **Yes: **Kenny, CHUI CHI MAN, PhD

---

## [Author Response · Author response to Decision Letter 0]

1 Aug 2021

Dear Professor,

The authors appreciate the opportunity to revise our manuscript entitled “The effect of a family-centered advance care planning intervention for persons with cognitive impairment and their family caregivers on end-of-life care discussions and decisions”. We have made extensive revisions to the manuscript thanks to the valuable comments and suggestions from both reviewers. 

To facilitate review, all suggested changes, as well as additional revisions to the manuscript, are in red font. We hope our manuscript is now acceptable for publication in PLOS ONE. Our responses are as the file of "response to reviewer".

---

## [Decision Letter · Decision Letter 1]

23 Aug 2022

The effect of a family-centered advance care planning intervention for persons with cognitive impairment and their family caregivers on end-of-life care discussions and decisions

PONE-D-21-14530R1

Dear Dr. Huang,

We’re pleased to inform you that your manuscript has been judged scientifically suitable for publication and will be formally accepted for publication once it meets all outstanding technical requirements.

Kind regards,

Khatijah Lim Abdullah, DClinP, MSc., BSc

Academic Editor

PLOS ONE

Additional Editor Comments (optional):

Dear Authors

Many thanks for your effort and time in revising the manuscript which is much clearer

However, there are some discrepancies and inconsistencies noted that need to be addressed :

Inconsistent wording: pre and post test or pre or post intervention (suggest the latter is better).

Inconsistent presentation: why p values in most tables but not in tables 7 and 8? Section 3.2.2: Please provide exact p-value for all variables.

Typos noted: PWCI vs PWIC

Proofreading needed

Reviewers' comments:

Reviewer's Responses to Questions

**Comments to the Author**

1. If the authors have adequately addressed your comments raised in a previous round of review and you feel that this manuscript is now acceptable for publication, you may indicate that here to bypass the “Comments to the Author” section, enter your conflict of interest statement in the “Confidential to Editor” section, and submit your "Accept" recommendation.

Reviewer #1: All comments have been addressed

Reviewer #3: All comments have been addressed

Reviewer #4: (No Response)

2. Is the manuscript technically sound, and do the data support the conclusions?

Reviewer #1: Partly

Reviewer #3: Yes

Reviewer #4: Yes

3. Has the statistical analysis been performed appropriately and rigorously? 

Reviewer #1: No

Reviewer #3: Yes

Reviewer #4: Yes

4. Have the authors made all data underlying the findings in their manuscript fully available?

Reviewer #1: Yes

Reviewer #3: Yes

Reviewer #4: Yes

5. Is the manuscript presented in an intelligible fashion and written in standard English?

Reviewer #1: No

Reviewer #3: Yes

Reviewer #4: No

---

## [Editor Report · Acceptance letter]

24 Aug 2022

PONE-D-21-14530R1 

The effect of a family-centered advance care planning intervention for persons with cognitive impairment and their family caregivers on end-of-life care discussions and decisions 

Dear Dr. Huang:

I'm pleased to inform you that your manuscript has been deemed suitable for publication in PLOS ONE. Congratulations! Your manuscript is now with our production department. 

Kind regards, 

on behalf of

Dr. Khatijah Lim Abdullah 

Academic Editor

PLOS ONE